# Emergency Management in the Event of Radiological Dispersion in an Urban Environment

**DOI:** 10.3390/s23042029

**Published:** 2023-02-10

**Authors:** Edoardo Cavalieri d’Oro, Andrea Malizia

**Affiliations:** 1Department of Industrial Engineering, University of Rome Tor Vergata, Via del Politecnico, 1, 00133 Rome, Italy; 2CBRN Unit and Laboratories of the Lombardy Region, Italian National Fire and Rescue Service, Via Messina 39, 20153 Milano, Italy; 3Department of Biomedicine and Prevention, University of Rome Tor Vergata, Via Montpellier, 1, 00133 Rome, Italy

**Keywords:** Radiological Dispersal Device (RDD), orphan sources, total effective dose (TED), first responders, emergency management

## Abstract

Dispersion of a radiological source is a complex scenario in terms of first response, especially when it occurs in an urban environment. The authors in this paper designed, simulated, and analyzed the data from two different scenarios with the two perspectives of an unintentional fire event and a Radiological Dispersal Device (RDD) intentional explosion. The data of the simulated urban scenario are taken from a real case of orphan sources abandoned in a garage in the center of the city of Milan (Italy) in 2012. The dispersion and dose levels are simulated using Parallel Micro Swift Spray (PMSS) software, which takes into account the topographic and meteorological information of the reference scenarios. Apart from some differences in the response system of the two scenarios analyzed, the information provided by the modeling technique used, compared to other models not able to capture the actual urban and meteorological contexts, make it possible to modulate a response system that adheres to the real impact of the scenario. The authors, based on the model results and on the evidence provided by the case study, determine the various countermeasures to adopt to mitigate the impact for the population and to reduce the risk factors for the first responders.

## 1. Introduction 

The risk scenarios where radioactive materials are being dispersed in the atmosphere include medical practices and industrial applications that use radioactive materials. The level of safety and security measures applied to radiological materials determines the relative likelihood that accidents occur, whether these are due to the improper applications of safety regulations (like abandoning a source instead of disposing it) or the inefficient custody of the source (which can lead to an illicit acquisition of the source by third parties).

This work is based on a real case that occurred in Milan in 2012 [1], where a former owner of a company performing radiological controls to the wings of small airplane declared to have abandoned the radiological sources he owned for his business before it went into bankruptcy, in a car garage in the center of Milan (Italy).

The scenarios are designed by the authors and the dispersion and dose levels are simulated using a Parallel Micro Swift Spray (PMSS) model with a horizontal resolution of 3 m, able to consider the deposition and dilution of radioactive materials following the profiles of the building characteristics of a selected urban scenario. The two cases analyzed and compared are those of an accidental fire and of a malevolent explosion of an RDD originating from abandoned radiological sources of Co-60 (Cobalt-60) and Cs-137 (Cesium-137). The results are expressed in terms of activity concentration in the air near the ground (Bq/m^3^), deposition on the ground (Bq/m^2^), and total effective dose (mSv) from inhalation and exposure to radiation. Based on the results, the authors determine the various countermeasures to adopt, as a function of distance from the source, and the actions to take to reduce risk factors for the population and the first responders. This study demonstrates, as do other studies in the literature [1,2,3,4,5,6,7,8,9,10], that wind and weather parameters remain key factors in assessing the dilution of radioactive materials, especially when in the evaluation it is possible to take into account the real geometry of the urban context considered. This means that the representation of the scenario obtained considering these variables affects the actions that need to be taken for the protection of the population and the first responders under emergency management plans. 

## 2. Scenario and Methods

### 2.1. Reference Scenario

Several computer code simulations are performed by verifying the possibility of structuring detailed analyses regarding the dispersion of radionuclides in urban areas and verifying the possible countermeasures that can be implemented to reduce the impact in terms of dose on the population and on rescuers [1,2,3,4,5,6,7,8,9,10].

Two distinct types of scenarios are evaluated and then compared with each other: the dispersion of radioactive materials involved due to an explosion and then to a fire. In the real case, the scenario did not lead to consequences due to a series of operations to ensure the safety of the environment in which the orphan sources were found and the subsequent controlled removal of the sources. Among the possible risk assessments performed at the time, some of the scenarios analyzed had predicted that a fire could break out in the garage or that a terrorist provokes intentionally an explosion in the garage causing the dispersion of radiological materials.

Both scenarios are considered plausible even today, 11 years after the event, because it must be considered that the garage of the discovery was full of flammable materials and that the most complex part of assembling an RDD is finding the radioactive materials, which were available beyond a door closed with a traditional lock.

The sources considered for the simulations are the two gamma emitters, Co-60 and Ce-137. The estimated activity at the time of discovery for the Co-60 source was 10 Giga Becquerel (GBq). This value was taken as a reference and used by the authors for both radionuclides for the first set of simulations; then, a second set of simulations tested the impact of higher activity sources. 

Although a 10 GBq source sounds modest in absolute terms, it is significant for the event (the discovery of unprotected orphan sources in the city center). It is possible to use a representation provided by the IAEA [2], summarized in Figure 1, to frame the dimension of the problem of a radiological accident from the point of view of the potential impact and the complexity of the countermeasures to be implemented.

The diagram in Figure 1 shows the various types of radiological incidents on a scale that classifies the potential impact of the accident scenarios using the activity of the source as an evaluation metric, as well as the type and place where it is located [2]. The scale of the impact caused by an accident involving such materials defines a quasi-proportional relationship between overall activity and potential impact. If, in addition, the source is involved in a fire or explosion, the precautionary measures to be taken to intervene further increase. For example, according to the IAEA, when a huge source (TBq) is involved in fire or explosion, a 300 m zoning must be maintained. Therefore, the discovery of orphan sources for industrial use, of activity of 10 GBq, can be framed as an average type of scenario and be related to a potential accident of medium consequences and medium complexity. Once the reference case is discussed, the activities of the two sources found are increased by a factor of 10,000 and 1000 respectively, changing the scale of complexity of the scenario. These kinds of high activity sources can be found in teletherapy applications.

### 2.2. Software Code

The simulation software used to recreate the two scenarios (fire and explosion) is the Micro SPRAY [3] software, a 3D Lagrangian particle code working at a microscale level, able to consider the presence of obstacles with a spatial resolution of 3 m. The software is combined with the SWIFT software [4], an analytically modified mass consistent interpolator over complex terrain, that combines the dispersion model with the information about the topography, the meteorological data, and the buildings of a preselected area. The wind data and weather conditions of a typical day used for the simulations were obtained from the database of the weather stations of the environmental agency of the Lombardy region (Italy) [5]. The data of the orography and the volumes of the buildings were acquired from the geoportal of the city of Milan [6]. In conclusion, compared to Gaussian-type models (generally used to simulate the dispersion and deposition of radionuclides), the SPRAY model allows considering detailed information that cannot be reproduced by software like Hotspot [7], such as
the position and geometry of the buildings that represent the urban context;the measured or expected dynamic and non-stationary meteorological and wind conditions, such as atmospheric classes, which however cannot take into account ground wind speed values lower than 1 m/s;calculation of the concentration of particulate matter also located in the vicinity of the point of origin (Gaussian models often see it as a point with infinite concentration);calculation of surface contamination resulting from dispersion even at different heights other than the ground and on vertical walls of buildings or obstacles.

Through this modeling system, it is therefore possible to evaluate the radiological consequences on the environment and on individuals who in any capacity (resident or transient population, and rescuers) find themselves stopping within the impact area of the scenario. 

The impact on the environment is assessed in terms of surface contamination from the deposition of radioactive particles measured in Bq/m^2^, while the dose received by individuals parked in the area where the plume passage is located is calculated in terms of the total effective dose. The total effective dose (TED) is given by the sum of the absorption contributions by inhalation and by radiation and expressed in Sievert (Sv).

The calculations are performed for a time equal to the maximum deposition time of the two clouds for the weather and wind conditions measured on the selected day. The total time of the simulations amounts to 120 min, which as we will see for the scenarios considered and for the emissivity of the radioactive sources considered is a sufficient time to deposit the particulate on the ground and to dissipate the cloud below the environmental background levels. Ground recombination effects were not considered. The background used in the simulations is that measured on average over a calendar year in the city of Milan and equal to 3.3 μSv/year.

### 2.3. Radiological Sources

The sources analyzed, in analogy with what was found, are two gamma emitters, Co-60 and Cesium 137, with an activity of 10 GBq, and then respectively increased to 10^4^ and 10^3^ factors. Table 1 reports the main characteristic of the sources. 

For the calculation of the quantities of radionuclides deposited on the ground, the conversion tables provided by Eckerman and Leggett [8], reported in Table 2, were used, while for the calculation of the effective dose from direct exposure from soil and cloud and from inhalation, calculated for only adults, the tables taken from the CEVaD manual [9] were used.

The presence of a seal is not included in the modeling of the source term, assuming it is destroyed or melted. In the real case, the sources were protected by an Ir-192 (Iridium-192) seal.

### 2.4. Emergency Management: Countermeasures to Protect the Population and Rescuers

The study analyzes two scenarios, one of safety and the other of security, namely the explosion of an RDD (an intentional event) and the outbreak of a fire (an accidental event). Our analysis focuses on issues related to the operations, role, and health status of the first responders involved in the first phase of the emergency, forgiving those aspects (e.g., communications, command, and control) that might mark the differences between safety and security scenarios.

The major goal for both scenarios is to protect the public and the first responders [11]. In relation to the information that models would make available during the response phase, the key figures in charge of the emergency intervention (i.e., the incident commander and the radiological assessor) should be informed as they make decisions related to the following:The definition of zones, disposition, and role of emergency personnel during the first response phase, recalling urgent technical assistance and rescue.The strategies and tactics aimed at limiting and optimizing the equivalent dose on the population and associated with the actions that the rescuers are called to carry out.The residence times (or stay time) in the various areas, into which the impacted area is divided, according to the allowable doses for the various categories of personnel dedicated to the emergency response.

The statuary forces appointed to intervene in a radiological emergency, or first responders, are divided into three main categories, according to the specific radiological risk to which each is allowed to be exposed: generic firefighters (FB-G), who are allowed a dose up to 20 mSv; firefighter specialists in NR emergencies (FB-NR), who are allowed up to 100 mSv; and a broader category of pure generic first responders, which includes police and medical personnel (FR-G), limited by a maximum exposure of 1 mSv. These dose limits are those referred to the Italian legislation implementing the Euratom European standard on the risks deriving from exposure to radioactive sources [12].

As defined by the IAEA radiological-nuclear emergency procedures [11], respiratory protection equipment must be worn within 100 m of a fire or explosion involving a potentially dangerous material. IAEA guidelines also specify that when very large sources (i.e., gamma sources with activities of the order of TBq) are involved in a fire, explosion, or fume, a 300 m radius must be kept, avoiding exposures that might result in severe deterministic effects. First responders, including police, firefighters, or other qualified emergency response personnel, have the duty to cordon off the inhabited areas resulting from such safety distances and ensure they are respected by the public. Figure 2 reports the inner and outer cordoned-off areas determined for the scenario based on the iso-dose profiles determined with the simulation software code; for simplicity in the representation, the map showing the urban settlements has been removed. Access to areas is restricted to classes of emergency workers allowed to be exposed to the radiation field that characterizes each zone. Moreover, the permanence should be justified by primary activities such as rescuing people in life-threatening situations, such as those who could be injured from the explosion, and by activities aimed to regain control of the situation, as this could extinguish the fire. It is also very important to support countermeasures in favor of securing the resident population, such as communicating recommendations (e.g., staying at home, closing windows, shutting off air recirculation systems), or assisting in safe evacuation from areas to the extent this is practicable.

In all areas, it is mandatory for emergency workers to use protective clothing and respiratory protection equipment to reduce the possibility of inhaling radioactive material by selecting the highest level of respiratory protection available.

In the specific case to which this study refers, the following hypotheses have been made. Rescuers called to respond to the event (either the fire or the explosion) are immediately alerted about the presence of radiological sources and will therefore wear protective equipment and dose meters from the start of their actions. The arrival time on site, for the general and specialist fire brigade teams, considering the real distribution of firehouses in the examined context, are respectively set at 5 and 10 min after the 112 calls. Both teams are equipped with air-fed breathing systems, with composite cylinders whose charge can vary between 20 and 50 min, depending on the stress conditions and on the type of action.

The information provided by the model, assuming that they are usable by the incident commander from the first moments of the intervention, is used to calculate, for the two scenarios, the following quantities: prediction of the extension of the areas to be prohibited based on the impact generated by the event, inhalation dose with and without breathing apparatus, residence time, and dose received within the exclusion areas for each of the three selected categories of operators.

The iso-dose areas resulting from the model are also used to calculate the avoidable dose that determines the levels of intervention to be undertaken to reduce the exposure of the resident population. The criteria used are those reported by the ICRP, that is, evaluating the indoor shelter option for effective dose values between 5 and 50 mSv, and the evacuation option for the range between 50 and 500 mSv.

It is worth noting that the values of the observed variables are time-varying; for the mathematical model, this is every 10 min. Accordingly, the protection strategy continually adapts during the response, as do the actions of control in the exclusion areas, whose boundaries may vary with the evolution of the scenario. Relevant to this matter is the transition occurring, during the 2 h simulation, between the phase of the passage of the radioactive plume and the phase of the resulting ground contamination from deposition.

## 3. Results

### 3.1. Assumptions about the Parameters Used in the Simulations

A series of simulations were performed with the SPRAY model, considering and varying some main parameters, as shown below.

Impact area: Two different impact areas are considered in the simulations: a detailed area, centered on the release point and with a width of 1.3 km × 1.3 km, and one centered on the part of the plume with the highest concentration, with a width of about 2 km × 2 km.Radionuclides involved: Co-60 and Cs-137 are the sources considered. The simulation does not include the case in which the two sources are combined.Initiating events: Two different types of emission scenarios are hypothesized, one of a malicious or terrorist nature and one of an accidental nature. The first event is characterized by an almost instantaneous release due to an RDD explosion with emissions distributed vertically and with an almost instantaneous release duration. The second release is from a fire located inside the condominium courtyard, with the thermal rising of the fumes, is hypothesized to have a release duration of about 10 min, including the arrival of firefighters, since they are located less than 2 km from the release point. The modest-sized fire was simulated with a fire load with a power of 1.8 MW.Times and meteorological situation: The events had a two-hour duration, starting from 9:00 a.m. of a preselected day (1 April 2022), characterized by conditions in the morning that change from neutral to unstable/convective when the wind blew in southbound from the release point. The meteorological boundary conditions (flow entering the computation domain) derive from simulations performed with a Weather Research and Forecasting [13] model at a horizontal resolution of 1 km.

The typical dimensions of the radioactive particulate for the two sources and released by the fire are estimated to be greater than 10 µm, in agreement with other sources in the literature [14,15]. The deposition rate considered, typical of particulate matter, is 4 cm/s. The deposited material is considered as laying on the ground over the two-hour period.

### 3.2. RDD Scenario

Two distinct detonations of a Co-60 with different activities were performed: a medium activity source, having 10 GBq activity (as in the case of the real discovery), and a high activity source, charged with the activity of 1.5 × 10^5^ GBq (that can be thought as the value the activity would have had if manufactured several decades earlier, or simply as the value of a high activity source used for teletherapy [10]). The projection model of the fragments due to the explosion was built in analogy with what is reported in the Hotspot software manual, with an initial blast for 5 s, assuming the entire release of the activity of the radiological source distributed along the vertical axis at a level of about 150 m from the ground. The fragmented particles are emitted vertically, distributed over four sources projected upward; the extension of their vertical center of gravity depends on the amount of explosive substance considered (10 kg of TNT).

The weather conditions for these simulations are those measured and recorded by the meteorological stations of the city of Milan on the morning of a day selected as a reference. The three-dimensional map of the buildings provided by Google Earth software presents the results from the simulations. The simulations are conducted in a 2 km^2^ city domain grid with a 3 m resolution while recording the meteorological changes every 10 min on a 1 km wide grid.

#### 3.2.1. Medium-Activity Co-60 Source Dispersion from Detonation

An explosion of the 10 GBq Co-60 source, assuming the explosive was placed in the warehouse where the orphan sources were originally found, would generate a blast cloud moving from the point of release towards the south of the city, according to the conditions forecast for the morning of the selected day. As shown in Figure 3, the cloud advances for the portion of the city affected by the event and therefore into the streets south of the release point between buildings that, on average, have a height of about 8 m. It also partially channels into some side streets off the main one, with an estimated average speed of about 1.4 m per second, and therefore at about one-third of the wind speed that day at higher altitudes (or that would be measured in the absence of the barriers represented by the buildings of the city).

The figure also shows the wind trend simulated by the mathematical model, in this case halfway through the simulation, at 10.00 a.m. of the designated day, showing an air velocity around the building in which release is assumed of between 1 and 2 m/s. These trends make it possible to consider the perturbations on the ground of little relevance and therefore to ignore the term of exposure from resuspension, which in any case for the simulation times used is not relevant because, in general, this factor is calculated for the subsequent phases of the emergency, namely those in which any measures of habitability of a contaminated area and its potential reclamation are taken into consideration [16].

Evaluations in terms of impact as a function of the distance from the point of release are also fundamental in determining the countermeasures to be implemented both for the resident population and for the rescuers. For this reason, six receptor points were selected downwind, of which the first two were close to the release point, one inside the courtyard of the building where the explosion took place, and one immediately placed out of it. The total equivalent dose (TED) for an individual inside one of the areas impacted by the explosion for a period of two hours is determined by adding the contributions from inhalation, direct radiation from the cloud, and indirect radiation from the ground (or ground shield effect), obtained by multiplying the contaminant concentrations in air and soil by appropriate conversion factors as a function of the specific radionuclide (values are provided by the CEVaD manual [9], consistently with homologous values reported by ICRP Publications in the Series 60/70).

The simulations conducted report that the cloud resulting from the explosion would have an area of 0.14 km^2^ for a downwind distance of 1 km and a perimeter of 2.2 km. The transit of the cloud from one end of this area to the other takes about 20 min, depositing contaminated material on the ground in about 50 min. In terms of the concentration of contaminants in the air, the cloud thins out to values comparable to environmental radioactivity in about 30 min from the explosion, partially depositing on the ground in the following 20 min.

This is reflected in the values of the total equivalent dose measured downwind to the source; the cloud rapidly disperses, and the significant contributions of dose can be those associated with emergency personnel or the public staying inside the courtyard. As can be seen from Table 3, the dominant contribution to the TED is given by the inhaled dose, with a small contribution given by the dose irradiated by the cloud, and with a negligible contribution of ground shine.

The highest values of dose resulting in points “0” and “1” (inside and outside the court) remain below the limit set for the population of 1 mSv. In addition, if ground shine irradiation coming from the vertical walls of the buildings was to be counted within the court, change to reach the 1 mSv, even in the case of a prolonged stay time, is very low, especially considering the short residence time of the radioactive cloud.

In conclusion, the medium activity dispersion scenario of a Co-60 source does not reveal to be of impact from a radiological standpoint; more significant are the damage caused by the explosion and the emotional impact it would generate.

#### 3.2.2. High-Activity Co-60 Source Dispersion from Detonation

To have a situation characterized by greater complexity, it is hypothesized that the discovery of the source took place about fifteen years earlier than in the real case and therefore with an activity of the source at the beginning of its life equal to 1.5 × 10^5^ GBq, shifting the complexity of the management of the intervention as identified in Figure 1. In this case, the equivalent dose values obtained in the simulation time of two hours and at one hour were used to calculate the residence times as a function of the distance from the source for the various categories of first responders and for the population.

The outcome of these calculations is shown in Table 4, from which it is possible to deduce the following recommendations for the management of the intervention. The residents of the building that houses the court in which the explosion is located, given the value of almost 100 mSv, must remain inside the building, keeping the windows closed while waiting to leave the building for decontamination. The same applies to the occupants of the surrounding buildings and therefore and up to distances of about 2 km, whereas you can see for point 5, keeping a window open would mean reaching values higher than 3 mSv. To ensure a continuous support function for the resident population in the court or to help any injured in the court, the help of specialized RN rescue teams is needed, because a generic team could already reach the limit value of 20 mSv in less than half an hour, to be exact, in 25 min inside the court. For safety and support operations for the population outside the court, it is possible to resort to the use of teams of generic rescuers, since the calculated times indicate the possibility of stopping for times between the almost two hours of point 1 and the 12 h measured 1769 km from the release point. Population evacuation measures were not considered, because they were considered too complex with respect to an exposure condition characterized by a preponderant risk from inhalation but of limited duration because it was less than 1 h.

### 3.3. High-Activity Cs-137 Source Dispersion from Fire

A further series of simulations was conducted assuming a fire, of an accidental rather than intentional nature, which could take place in the warehouse that, in addition to the radioactive sources, contained large quantities of combustible and flammable materials. In this case, the source is composed of Cs-137 with a high activity of 2 × 10^3^ GBq. The following data were considered in constructing this further scenario: thermal power of the initial fire of 1800 kW, generated by the presence of 15 kg of propane, 5 kg of gasoline, 5 kg of plastic, and 5 kg of wood in the approximately 50 square meters of the warehouse, considering 5% of cold fumes and a duration of a few tens of minutes before extinguished by the fire brigade.

Since Cs-137, even in its most fire-resistant compositions, such as for example in the oxide format, melts at much lower temperatures than Co-60, as also expressed by its fire release fraction [2], it lends itself better to fire simulation. However, compared to the detonation scenario, a series of differences is expressed by the simulations. First, the concentration of activity in the air of the cloud along the wind direction does not show a decreasing behavior as a function of distance, as was the case for the explosion. This is due presumably to the combined effect of the hot and cold fumes of the fire. The cold part of the fumes hits the area near the point of release, and the hot part goes back up, bringing a maximum value at a certain distance from the point of emission. In addition, in the event of a fire, as reported in Table 5, the dispersion effect in the atmosphere seems to be higher, where an appreciable total equivalent dose value can be found in the vicinity of the source inside the courtyard of the building where the fire originates. In the remaining part of the downwind axis, the effects of the fire decay very rapidly with distance. The maximum value obtained from the fire simulations at each receptor point is calculated approximately 30 min from the start of the event.

By looking at simulations and considering that the fire burns for 10 min and the cold and hot fumes disperse in about 20 min, the average dose value of 12 mSv/h corresponds to about 6 mSv absorbed inside the courtyard in a 30-min period. This defines the dose potentially absorbed by inhalation for the inhabitants of the building facing the internal courtyard where the release source is located, assuming they do not take precautions such as closing windows. On the other hand, the event lasts for such a short time that it does not allow time to provide communications on how to protect oneself. The dose to which generic fire brigades intervening to extinguish the fire are exposed can be somewhat hypothetical or practically nil, as protected with self-contained breathing masks.

### 3.4. Scenario Comparison: Fire vs. Blast

Figure 4 shows the comparison between the dispersion provoked by the two simulated events.

Comparing the representations of the two columns of smoke and debris released and simulated by the model in the two scenarios, reported in Table 6, it is possible to observe the slightly wavy behavior of the fumes compared to the bottom-up scenario shown by the explosion case.

The cloud of the fire, due to the different densities and therefore weight, alternates in behavior, first rising, then descending, and finally dispersing over a larger distance. In large part, this occurs because the fire is extinguished in a short time, therefore damping its potential convective force, and before all the combustible material can release the source of radioactive material into the air. The case of the explosion, on the other hand, completely exhausts its source term by projecting it entirely to about 150 m of cloud height, where the wind blows at a speed about 2.5 times that of the one measured on ground. Considering the very similar activity values of the two sources (Co-60 vs. Cs-137), these observations explain both the greater diffusivity of the blast cloud compared to the fire cloud, and its greater concentration in the air, then reflected on the ground.

## 4. Conclusions

In this work, the authors tested and practiced the properties of a mathematical model capable of calculating and representing the concentrations in the air and on the ground resulting from the detonation or fire of radioactive sources in an urban environment. The model is a Lagrangian Particle Dispersion Model developed by ARIA Technologies, called ‘PMSS’ for Parallel Micro Swift Spray, validated to model the dispersion of radionuclides originating from a fire or explosion as reported from the most recent IAEA reports [17]. The results obtained with this model consider the weather conditions recorded or forecast over time and the urban context, that is, the replication of buildings and infrastructure, with their volumes. The software’s capabilities make it a useful tool for emergency planning, response, and recovery, allowing the visualization of the dispersion of hazardous materials on the ground and in the air on 3D maps that can be used also on georeferencing tools like Google Maps or QGis. The way the software visualizes information related to radiological releases in emergencies and associated uncertainties is of extreme importance for decision making, both for experts involved in emergency management and for the potentially affected population [18]. All these features would be sufficient to justify its adoption with respect to standard Gaussian models because it better suits the needs of representation that the decision- makers in charge of rescue operations might need to approach radiological emergencies. However, to these first observations, we can also add some findings concerning the exploration of the scenarios conducted. The simulations conducted take their cue from a real case of abandonment of sources (i.e., orphan sources) within a courtyard located in a densely populated urban center. This event occurred between 2012 and 2013 in the City of Milan in the north of Italy. The nuclear materials and the data used for the simulations, except for the meteorological situation, are taken from this real case. Then cases with higher activity were analyzed both to increase the level of complexity of the scenarios and in continuity with previous studies about these topics [10]. Several software simulations were performed to test the model capabilities, focusing on the differences in scenarios generated by different radiological sources, at a given weather condition; three of them are reported in this work to compare the outcomes in terms of the dispersions originating from a fire and from an RDD. The results obtained from the simulations, if compared with software with lower characteristics and described by greater uncertainty in the quantification of concentrations and doses, provide more easy-to-use information in favor of the decisions of the incident commander or to an expert analyst. Of relevance is the restriction of the intervention and exclusion areas, usually based on concentric representations of iso-dose lines generally overestimated by Gaussian models. The detailed representation provided by the PMSS software, while requiring a familiarization effort, leads to the determination of impact areas sized to the real impact of the phenomenon, thus leading to an efficiency of the resources to be allocated on the field. This, for instance, includes the option of visualizing the channeling effects of radioactive particles into main roads and city arteries, as well as that of ascertaining that the passage times of the cloud before it disperses at concentration levels comparable to the background are in fact shorter than what are generally perceived. Other features include the correct determination of quantities such as “concentration” in proximity of the source point, which Gaussian models have difficulty managing. In addition, the representation throughout colored bands that report dose ranges, versus the curves with a single dose value (iso-dose) provided by other software, seem appropriate to manage models’ uncertainties [18]. Finally, considering that the net computational time of some settings, for each of the simulations, was performed in less than 10 min on a simple laptop, make the software suitable to be used for the real-time management of complex RN emergencies.

Future developments of this work envision performing calculations with other simulation software and management tools, such as Hotspot or NBC Analysis, and to verify the correspondence of the results with those provided in this study.

## Figures and Tables

**Figure 1 sensors-23-02029-f001:**
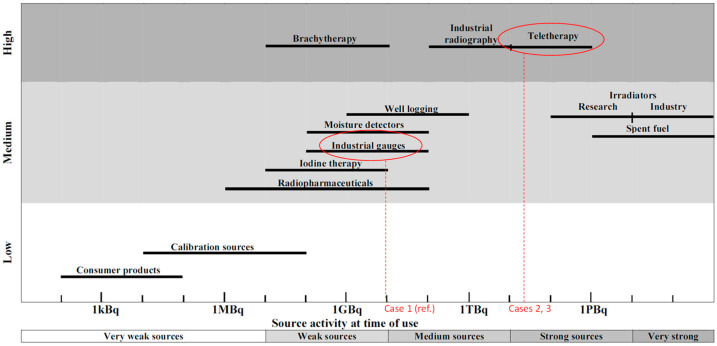
A classification that compares the severity and complexity of the different incidents as a function of the activity. This figure is readapted from IAEA 1162 [2].

**Figure 2 sensors-23-02029-f002:**
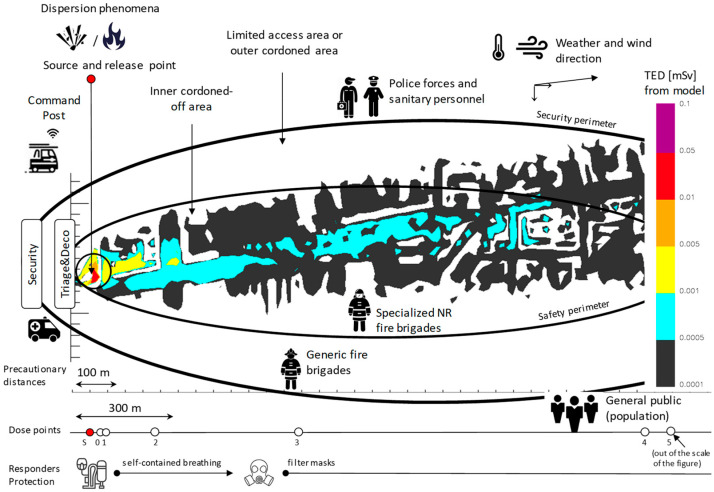
Representation of the areas of respect, of the protection measures to be respected in each area, and of the resources allocated by area between the entities of an RN (radiological-nuclear) emergency response.

**Figure 3 sensors-23-02029-f003:**
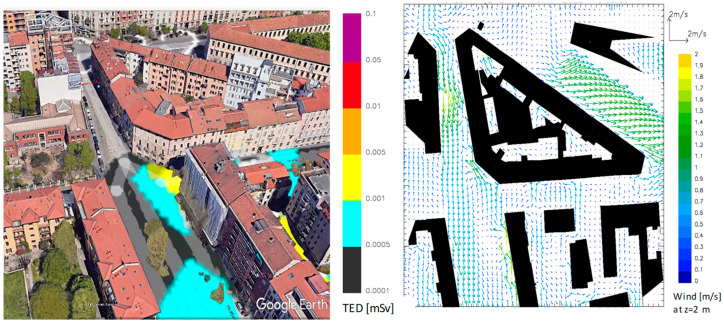
3D representation of the contamination of the area and 2D scheme reporting wind turbulence around the same building at a level between 0 and 2 m above the ground.

**Figure 4 sensors-23-02029-f004:**
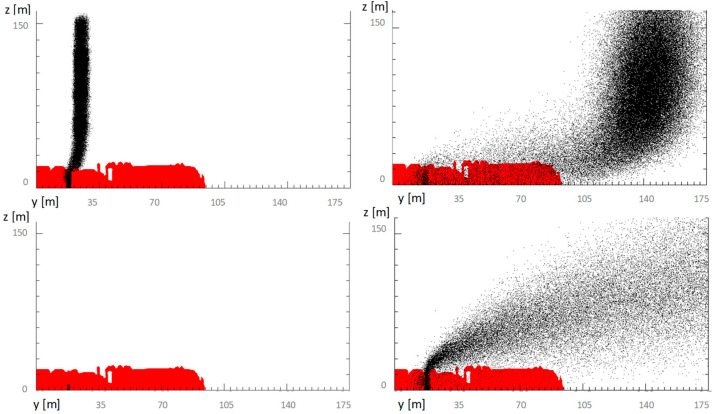
Column of particulate matter from explosion between 10 s and 150 s (in red) vs. cloud of smoke from the fire between 5 s and 10 min (in black).

**Table 1 sensors-23-02029-t001:** Main properties of the two radionuclides considered in the scenarios.

Radionuclides	T 1/2 [Years]	Energy * [MeV]	Melting Point (+) [°C]	Activities [Bq]	
Co-60	5.27 years	1.1732	1495	1.00 × 10^10^	1.50 × 10^14^
Cs-137	30.17 years	0.6617	490	1.00 × 10^10^	1.90 × 10^13^

* Gamma energy of the most probable decay. + Pure cesium melts at 28.5 degrees, but the table reports cesium oxide.

**Table 2 sensors-23-02029-t002:** Direct radiation from the plume and from the ground.

Radionuclide	Effective Dose Coefficient		Effective Dose Rate per Unit of Deposition on the Ground	
	Plume [Sv/h]	Ground [Bq/m^2^]	Plume [Sv/h]	Ground [Bq/m^2^]
Co-60	1.2 × 10^−13^	2.3 × 10^−15^	4.3 × 10^−10^	8.3 × 10^−12^
Cs-137 *	2.6 × 10^−14^	5.5 × 10^−16^	9.4 × 10^−11^	2.0 × 10^−12^

* The dose coefficients for the radionuclide include the contribution of the decay products.

**Table 3 sensors-23-02029-t003:** Contributions to the total equivalent dose originating from the detonation of a 10 GBq Co-60 source.

	Point 0 (in)	Point 1 (out)	Point 2	Point 3	Point 4	Point 5
Distance from Source [m]	26	24	164	645	950	1769
TED [mSv]	6.18	1.41	1.07	0.55	0.5	0.23
Inhaled Dose [mSv]	6.06	1.38	1.05	0.53	0.49	0.22
Direct Dose [mSv]	0.09	0.02	0.02	0.01	0.01	0.00
Indirect Dose [mSv]	3.64 × 10^−2^	7.81 × 10^−3^	5.15 × 10^−3^	3.09 × 10^−3^	3.09 × 10^−3^	5.01 × 10^−4^

**Table 4 sensors-23-02029-t004:** Calculation of the equivalent dose and of the average residence time at various distances from the point of explosion of a source of Co-60 with an activity of 1.5 × 10^5^ GBq.

	Units	Point_0 (in)	Point_1 (out)	Point_2	Point_3	Point_4	Point_5
Distance	meters	26	24	164	645	950	1769
TED (h)	mSv/h	46.4	10.6	8.0	4.1	3.8	1.7
Tstay POP	minutes	1	6	8	15	16	35
Tstay FR-G	hours, minutes	25 m	1 h 53 m	2 h 30 m	4 h 53 m	5 h 17 m	11 h 41 m
Tstay FB-NR	hours, minutes	2 h 10 m	9 h 28 m	12 h 30 m	24 h 27 m	26 h 27 m	58 h 28 m

**Table 5 sensors-23-02029-t005:** Calculation of dose and residence time at various distances from the fire point of a source of Cs-137 with an activity of 1.9 × 10^3^ GBq, calculated for its maximum (20 min after ignition).

	Units	Point_0	Point_1	Point_2	Point_3	Point_4	Point_5
Distance	meters	26	24	164	645	950	1769
TED (h)	mSv/h	12.0	0.4	0.2	0.03	0.1	0.1

**Table 6 sensors-23-02029-t006:** Comparison of the two simulations (detonation vs. fire), reporting the shape and concentration values of the radioactive clouds.

Scenario	Type of Data	Units	Point_0	Point_1	Point_2	Point_3	Point_4	Point_5
RDD	Air Concentration	Bq/m^3^	4.02 × 10^4^	5.54 × 10^3^	5.07 × 10^4^	1.47 × 10^5^	3.02 × 10^5^	5.63 × 10^5^
Co-60 High	Ground Deposition	Bq/m^2^	3.59 × 10^8^	7.69 × 10^7^	5.10 × 10^7^	3.07 × 10^7^	3.60 × 10^7^	5.33 × 10^6^
Fire	Air Concentration	Bq/m^3^	1.04 × 10^3^	1.00 × 10^4^	2.04 × 10^4^	7.79 × 10^1^	2.22 × 10^3^	5.27 × 10^3^
Cs-137 High	Ground Deposition	Bq/m^2^	7.12 × 10^7^	2.36 × 10^6^	1.34 × 10^6^	2.47 × 10^5^	8.96 × 10^3^	2.84 × 10^5^

## Data Availability

Not applicable.

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
