# Peer review of "Emergency Management in the Event of Radiological Dispersion in an Urban Environment"

_sensors, 2023, doi:10.3390/s23042029_

Round 1

Reviewer 1 Report

The paper must be revived on terminology end uniformization of terms/symbols etc., definitions (you use “GBq”, “GB”, “Gbq”; “cobalt-60”, “Co-60”, “cobalt 60” … etc.,) used along the text.

Good paper.

Reviewer 2 Report

The article „Emergency management in the event of radiological dispersion in the urban environment“ deals with hypothetical situations when Cs-137 or Co-60 is dispersed via explosion or combustion. Two levels of activity scenarios are considered; GBq sources and hundreds of TBq sources. The activities were used due to a real scenario, where two RN sources (Cs-137 and Co-60) were discovered in the abandoned facility. The management of rescue is based on the prediction of  Micro SPRAY and SWIFT SW calculation. The language is appropriate and the manuscript is divided according to the manuscript layout. The references are relevant and in reasonable numbers.

However, some issues have to be clarified before publication:

1.       Use only dots as decimal points (e.g. Tab. 1 and 2)

2.       Tab. 1 – Co-60 doesn’t have a decay energy of 11.732 MeV, has 1173 keV and 1332 with each yield of almost 100%, so at one decay, these two photons are emitted. There is also a sum of these lines, 2.5 MeV, but low probability.

3.       When stating, according to IAEA, that very large sources (TBq) are employed, a 300m circle has to be kept. The right description is based on the dose rate since these sources can have low photons energy (Am-241) or be shielded. Therefore, employing some measures are meaningless based only on activity.

4.       Do not use the dirty bomb term. This term is used by people who lack a broader understanding of CBRN problematics – use explosive RDD.

5.       Tab. 6 – there is also a picture without a caption. Please separate the picture from the table and add an axis to the picture and description.

6.       A more serious issue that has to be addressed is, that the whole manuscript doesn’t count with the sealed radionuclide source, as it is stated in the beginning. All industrial sources are sealed in special stainless steel containers (2 layers) and inside the capsule, there is ceramic, which consists of the radionuclide. Therefore the hypothesis of fire or detonation is off-topic since only a small fraction of RN could be released if any. In case of an explosion, the capsule will behave as a bullet and there is certainly a higher probability that it kill someone than the radionuclide does.

7.       The second serious issue is, that there is prediction only from one SW. These SWs are not reliable, especially in urban areas due to many aspects. There should be the comparison of other SW predictions, such as HPAC or NBC analysis.

Based on the above stated, I recommend major revisions.

Reviewer 3 Report

1. In the abstract, please check the superscript of (Bq / m3) and (Bq / m2). hte same comments in the text

2. There are 9 keywords, usually any paper contains 5-6 keywords only

3. The introduction is very short, and contains ONLY 1 reference, you need to extend it and show the importance of this work and you need to add some recent Refs

4. First line in subsection (2.1): Several computer code simulations are performed by verifying the possibility of structuring detailed analyzes regarding.....Please add some references to this sentence 

5. Please improve the resolution of Fig1

6. After Fig.1: please add space in this sentence:  Figure 1shows.........

7. In second line in section 2.3: with an activity of 10 Gbq....is this GBq??

8. In Table 1, is the energy of Co-60: 11.732 MeV? or 1.732 MeV. please check

9. I dont understand why the authors add figure in Table 6?

10. The conclusion is very long, you can shorten it and focus on the main findings in this work
